# Latent Maximum Margin Clustering

**Guang-Tong Zhou, Tian Lan, Arash Vahdat, and Greg Mori**
School of Computing Science
Simon Fraser University
{gza11,tla58,avahdat,mori}@cs.sfu.ca

## Abstract

We present a maximum margin framework that clusters data using latent variables. Using latent representations enables our framework to model unobserved information embedded in the data. We implement our idea by large margin learning, and develop an alternating descent algorithm to effectively solve the resultant non-convex optimization problem. We instantiate our latent maximum margin clustering framework with tag-based video clustering tasks, where each video is represented by a latent tag model describing the presence or absence of video tags. Experimental results obtained on three standard datasets show that the proposed method outperforms non-latent maximum margin clustering as well as conventional clustering approaches.

## 1 Introduction

Clustering is a major task in machine learning and has been extensively studied over decades of research [11]. Given a set of observations, clustering aims to group data instances of similar structures or patterns together. Popular clustering approaches include the k-means algorithm [7], mixture models [22], normalized cuts [27], and spectral clustering [18]. Recent progress has been made using maximum margin clustering (MMC) [32], which extends the supervised large margin theory (e.g. SVM) to the unsupervised scenario. MMC performs clustering by simultaneously optimizing cluster-specific models and instance-specific labeling assignments, and often generates better performance than conventional methods [33, 29, 37, 38, 16, 6].

Modeling data with latent variables is common in many applications. Latent variables are often defined to have intuitive meaning, and are used to capture unobserved semantics in the data. As compared with ordinary linear models, latent variable models feature the ability to exploit a richer representation of the space of instances. Thus, they often achieve superior performance in practice. In computer vision, this is best exemplified by the success of deformable part models (DPMs) [5] for object detection. DPMs enhance the representation of an object class by capturing viewpoint and pose variations. They utilize a root template describing the entire object appearance and several part templates. Latent variables are used to capture deformations and appearance variations of the root template and parts. DPMs perform object detection via search for the best locations of the root and part templates.

Latent variable models are often coupled with supervised learning to learn models incorporating the unobserved variables. For example, DPMs are learned in a latent SVM framework [5] for object detection; similar models have been shown to improve human action recognition [31]. A host of other applications of latent SVMs have obtained state-of-the-art performance in computer vision. Motivated by their success in supervised learning, we believe latent variable models can also help in unsupervised clustering – data instances with similar latent representations should be grouped together in one cluster.

As the latent variables are unobserved in the original data, we need a learning framework to handle this latent knowledge. To implement this idea, we develop a novel clustering algorithm based on MMC that incorporates latent variables – we call this latent maximum margin clustering (LMMC). The LMMC algorithm results in a non-convex optimization problem, for which we introduce an

iterative alternating descent algorithm. Each iteration involves three steps: inferring latent variables for each sample point, optimizing cluster assignments, and updating cluster model parameters.

To evaluate the efficacy of this clustering algorithm, we instantiate LMMC for tag-based video clustering, where each video is modeled with latent variables controlling the presence or absence of a set of descriptive tags. We conduct experiments on three standard datasets: TRECVID MED 11 [19], KTH Actions [26] and UCF Sports [23], and show that LMMC outperforms non-latent MMC and conventional clustering methods.

The rest of this paper is organized as follows. Section 2 reviews related work. Section 3 formulates the LMMC framework in detail. We describe tag-based video clustering in Section 4, followed by experimental results reported in Section 5. Finally, Section 6 concludes this paper.

## 2 Related Work

**Latent variable models**. There has been much work in recent years using latent variable models. The definition of latent variables are usually task-dependent. Here we focus on the learning part only. Andrews et al. [1] propose multiple-instance SVM to learn latent variables in positive bags. Felzenszwalb et al. [5] formulate latent SVM by extending binary linear SVM with latent variables. Yu and Joachims [36] handle structural outputs with latent structural SVM. This model is also known as maximum margin hidden conditional random fields (MMHCRF) [31]. Kumar et al. [14] propose self-paced learning, an optimization strategy that focuses on simple models first. Yang et al. [35] kernelize latent SVM for better performance. All of this work demonstrates the power of max-margin latent variable models for supervised learning; our framework conducts unsupervised clustering while modeling data with latent variables.

**Maximum margin clustering**. MMC was first proposed by Xu et al. [32] to extend supervised large margin methods to unsupervised clustering. Different from the supervised case, where the optimization is convex, MMC results in non-convex problems. To solve it, Xu et al. [32] and Valizadegan and Rong [29] reformulate the original problem as a semi-definite programming (SDP) problem. Zhang et al. [37] employ alternating optimization – finding labels and optimizing a support vector regression (SVR). Li et al. [16] iteratively generate the most violated labels, and combine them via multiple kernel learning. Note that the above methods can only solve binary-cluster clustering problems. To handle the multi-cluster case, Xu and Schuurmans [33] extends the SDP method in [32]. Zhao et al. [38] propose a cutting-plane method which uses the constrained convex-concave procedure (CCCP) to relax the non-convex constraint. Gopalan and Sankaranarayanan [6] examine data projections to identify the maximum margin. Our framework deals with multi-cluster clustering, and we model data instances with latent variables to exploit rich representations. It is also worth mentioning that MMC leads naturally to the semi-supervised SVM framework [12] by assuming a training set of labeled instances [32, 33]. Using the same idea, we could extend LMMC to semi-supervised learning.

MMC has also shown its success in various computer vision applications. For example, Zhang et al. [37] conduct MMC based image segmentation. Farhadi and Tabrizi [4] find different view points of human activities via MMC. Wang and Cao [30] incorporate MMC to discover geographical clusters of beach images. Hoai and Zisserman [8] form a joint framework of maximum margin classification and clustering to improve sub-categorization.

**Tag-based video analysis**. Tagging videos with relevant concepts or attributes is common in video analysis. Qi et al. [20] predict multiple correlative tags in a structural SVM framework. Yang and Toderici [34] exploit latent sub-categories of tags in large-scale videos. The obtained tags can assist in recognition. For example, Liu et al. [17] use semantic attributes (e.g. up-down motion, torso motion, twist) to recognize human actions (e.g. walking, hand clapping). Izadinia and Shah [10] model low-level event tags (e.g. people dancing, animal eating) as latent variables to recognize complex video events (e.g. wedding ceremony, grooming animal).

Instead of supervised recognition of tags or video categories, we focus on unsupervised tag-based video clustering. In fact, recently research collects various sources of tags for video clustering. Schroff et al. [25] cluster videos by the capturing locations. Hsu et al. [9] build hierarchical clustering using user-contributed comments. Our paper uses latent tag models, and our LMMC framework is general enough to handle various types of tags.

# 3 Latent Maximum Margin Clustering

As stated above, modeling data with latent variables can be beneficial in a variety of supervised applications. For unsupervised clustering, we believe it also helps to group data instances based on latent representations. To implement this idea, we propose the LMMC framework.

LMMC models instances with latent variables. When fitting an instance to a cluster, we find the optimal values for latent variables and use the corresponding latent representation of the instance. To best fit different clusters, an instance is allowed to flexibly take different latent variable values when being compared to different clusters. This enables LMMC to explore a rich latent space when forming clusters. Note that in conventional clustering algorithms, an instance is usually restricted to have the same representation in all clusters. Furthermore, as the latent variables are unobserved in the original data, we need a learning framework to exploit this latent knowledge. Here we develop a large margin learning framework based on MMC, and learn a discriminative model for each cluster. The resultant LMMC optimization is non-convex, and we design an alternating descent algorithm to approximate the solution. Next we will briefly introduce MMC in Section 3.1, followed by detailed descriptions of the LMMC framework and optimization respectively in Sections 3.2 and 3.3.

## 3.1 Maximum Margin Clustering

MMC [32, 37, 38] extends the maximum margin principle popularized by supervised SVMs to unsupervised clustering, where the input instances are unlabeled. The idea of MMC is to find a labeling so that the margin obtained would be maximal over all possible labelings. Suppose there are $N$ instances $\{\mathbf{x}_i\}_{i=1}^N$ to be clustered into $K$ clusters, MMC is formulated as follows [33, 38]:

$$\min_{\mathcal{W}, \mathcal{Y}, \xi \geq 0} \quad \frac{1}{2} \sum_{t=1}^K ||\mathbf{w}_t||^2 + \frac{C}{K} \sum_{i=1}^N \sum_{r=1}^K \xi_{ir} \tag{1}$$

$$\text{s.t.} \quad \sum_{t=1}^K y_{it} \mathbf{w}_t^\top \mathbf{x}_i - \mathbf{w}_r^\top \mathbf{x}_i \geq 1 - y_{ir} - \xi_{ir}, \; \forall i, r$$

$$y_{it} \in \{0, 1\}, \; \forall i, t \qquad \sum_{t=1}^K y_{it} = 1, \; \forall i$$

where $\mathcal{W} = \{\mathbf{w}_t\}_{t=1}^K$ are the linear model parameters for each cluster, $\xi = \{\xi_{ir}\}$ ($i \in \{1, \ldots, N\}$, $t \in \{1, \ldots, K\}$) are the slack variables to allow soft margin, and $C$ is a trade-off parameter. We denote the labeling assignment by $\mathcal{Y} = \{y_{it}\}$ ($i \in \{1, \ldots, N\}$, $t \in \{1, \ldots, K\}$), where $y_{it} = 1$ indicates that the instance $\mathbf{x}_i$ is clustered into the $t$-th cluster, and $y_{it} = 0$ otherwise. By convention, we require that each instance is assigned to one and only one cluster, i.e. the last constraint in Eq. 1. Moreover, the first constraint in Eq. 1 enforces a large margin between clusters by constraining that the score of $\mathbf{x}_i$ to the assigned cluster is sufficiently larger than the score of $\mathbf{x}_i$ to any other clusters. Note that MMC is an unsupervised clustering method, which jointly estimates the model parameters $\mathcal{W}$ and finds the best labeling $\mathcal{Y}$.

**Enforcing balanced clusters**. Unfortunately, solving Eq. 1 could end up with trivial solutions where all instances are simply assigned to the same cluster, and we obtain an unbounded margin. To address this problem, we add cluster balance constraints to Eq. 1 that require $\mathcal{Y}$ to satisfy

$$L \leq \sum_{i=1}^N y_{it} \leq U, \; \forall t \tag{2}$$

where $L$ and $U$ are the lower and upper bounds controlling the size of a cluster. Note that we explicitly enforce cluster balance using a hard constraint on the cluster sizes. This is different from [38], a representative multi-cluster MMC method, where the cluster balance constraints are implicitly imposed on the accumulated model scores (i.e. $\sum_{i=1}^N \mathbf{w}_t^\top \mathbf{x}_i$). We found empirically that explicitly enforcing balanced cluster sizes led to better results.

## 3.2 Latent Maximum Margin Clustering

We now extend MMC to include latent variables. The latent variable of an instance is cluster-specific. Formally, we denote $\mathbf{h}$ as the latent variable of an instance $\mathbf{x}$ associated to a cluster parameterized by $\mathbf{w}$. Following the latent SVM formulation [5, 36, 31], scoring $\mathbf{x}$ w.r.t. $\mathbf{w}$ is to solve an

inference problem of the form:

$$f_{\mathbf{w}}(\mathbf{x}) = \max_{\mathbf{h}} \mathbf{w}^{\top} \Phi(\mathbf{x}, \mathbf{h}) \tag{3}$$

where $\Phi(\mathbf{x}, \mathbf{h})$ is the feature vector defined for the pair of $(\mathbf{x}, \mathbf{h})$. To simplify the notation, we assume the latent variable $\mathbf{h}$ takes its value from a discrete set of labels. However, our formulation can be easily generalized to handle more complex latent variables (e.g. graph structures [36, 31]).

To incorporate the latent variable models into clustering, we replace the linear model $\mathbf{w}^{\top}\mathbf{x}$ in Eq. 1 by the latent variable model $f_{\mathbf{w}}(\mathbf{x})$. We call the resultant framework latent maximum margin clustering (LMMC). LMMC finds clusters via the following optimization:

$$\min_{\mathcal{W}, \mathcal{Y}, \xi \geq 0} \quad \frac{1}{2} \sum_{t=1}^{K} ||\mathbf{w}_t||^2 + \frac{C}{K} \sum_{i=1}^{N} \sum_{r=1}^{K} \xi_{ir} \tag{4}$$

$$\text{s.t.} \quad \sum_{t=1}^{K} y_{it} f_{\mathbf{w}_t}(\mathbf{x}_i) - f_{\mathbf{w}_r}(\mathbf{x}_i) \geq 1 - y_{ir} - \xi_{ir}, \ \forall i, r$$

$$y_{it} \in \{0, 1\}, \ \forall i, t \quad \sum_{t=1}^{K} y_{it} = 1, \ \forall i \quad L \leq \sum_{i=1}^{N} y_{it} \leq U, \ \forall t$$

We adopt the notation $\mathcal{Y}$ from the MMC formulation to denote the labeling assignment. Similar to MMC, the first constraint in Eq. 4 enforces the large margin criterion where the score of fitting $\mathbf{x}_i$ to the assigned cluster is marginally larger than the score of fitting $\mathbf{x}_i$ to any other clusters. Cluster balance is enforced by the last constraint in Eq. 4. Note that LMMC jointly optimizes the model parameters $\mathcal{W}$ and finds the best labeling assignment $\mathcal{Y}$, while inferring the optimal latent variables.

### 3.3 Optimization

It is easy to verify that the optimization problem described in Eq. 4 is non-convex due to the optimization over the labeling assignment variables $\mathcal{Y}$ and the latent variables $\mathcal{H} = \{\mathbf{h}_{it}\}$ ($i \in \{1, \dots, N\}, t \in \{1, \dots, K\}$). To solve it, we first eliminate the slack variables $\xi$, and rewrite Eq. 4 equivalently as:

$$\min_{\mathcal{W}} \frac{1}{2} \sum_{t=1}^{K} ||\mathbf{w}_t||^2 + \frac{C}{K} R(\mathcal{W}) \tag{5}$$

where $R(\mathcal{W})$ is the risk function defined by:

$$R(\mathcal{W}) \quad = \quad \min_{\mathcal{Y}} \sum_{i=1}^{N} \sum_{r=1}^{K} \max \left(0, 1 - y_{ir} + f_{\mathbf{w}_r}(\mathbf{x}_i) - \sum_{t=1}^{K} y_{it} f_{\mathbf{w}_t}(\mathbf{x}_i)\right) \tag{6}$$

$$\text{s.t.} \quad y_{it} \in \{0, 1\}, \ \forall i, t \quad \sum_{t=1}^{K} y_{it} = 1, \ \forall i \quad L \leq \sum_{i=1}^{N} y_{it} \leq U, \ \forall t$$

Note that Eq. 5 minimizes over the model parameters $\mathcal{W}$, and Eq. 6 minimizes over the labeling assignment variables $\mathcal{Y}$ while inferring the latent variables $\mathcal{H}$. We develop an alternating descent algorithm to find an approximate solution. In each iteration, we first evaluate the risk function $R(\mathcal{W})$ given the current model parameters $\mathcal{W}$, and then update $\mathcal{W}$ with the obtained risk value. Next we describe each step in detail.

**Risk evaluation**: The first step of learning is to compute the risk function $R(\mathcal{W})$ with the model parameters $\mathcal{W}$ fixed. We first infer the latent variables $\mathcal{H}$ and then optimize the labeling assignment $\mathcal{Y}$. According to Eq. 3, the latent variable $\mathbf{h}_{it}$ of an instance $\mathbf{x}_i$ associated to cluster $t$ can be obtained via: $\text{argmax}_{\mathbf{h}_{it}} \mathbf{w}_t^{\top} \Phi(\mathbf{x}_i, \mathbf{h}_{it})$. Note that the inference problem is task-dependent. For our latent tag model, we present an efficient inference method in Section 4.

After obtaining the latent variables $\mathcal{H}$, we optimize the labeling assignment $\mathcal{Y}$ from Eq. 6. Intuitively, this is to minimize the total risk of labeling all instances yet maintaining the cluster balance constraints. We reformulate Eq. 6 as an integer linear programming (ILP) problem by introducing a variable $\psi_{it}$ to capture the risk of assigning an instance $\mathbf{x}_i$ to a cluster $t$. The ILP can be written as:

$$R(\mathcal{W}) = \min_{\mathcal{Y}} \sum_{i=1}^{N} \sum_{t=1}^{K} \psi_{it} y_{it} \quad \text{s.t.} \ y_{it} \in \{0, 1\}, \forall i, t \quad \sum_{t=1}^{K} y_{it} = 1, \forall i \quad L \leq \sum_{i=1}^{N} y_{it} \leq U, \forall t \tag{7}$$

| | **Cluster**: feeding animal | | | | | | | | **Cluster**: board trick | | | | | | | | |
|---|---|---|---|---|---|---|---|---|---|---|---|---|---|---|---|---|---|
| video | 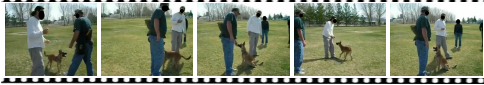 | | | | | | | |  | | | | | | | | |
| $\mathcal{T}$: | board | car | dog | food | grass | man | snow | tree $\cdots$ | board | car | dog | food | grass | man | snow | tree $\cdots$ |
| **h**: | 0 | 0 | 1 | 1 | 1 | 1 | 0 | 1 $\cdots$ | 1 | 0 | 0 | 0 | 1 | 1 | 0 | 0 $\cdots$ |

Figure 1: Two videos represented by the latent tag model. Please refer to the text for details about $\mathcal{T}$ and **h**. Note that the cluster labels (i.e. "feeding animal", "board trick") are unknown beforehand. They are added for a better understanding of the video content and the latent tag representations.

where $\psi_{it} = \sum_{r=1, r\neq t}^{K} \max(0, 1 + f_{\mathbf{w}_r}(\mathbf{x}_i) - f_{\mathbf{w}_t}(\mathbf{x}_i))$. This captures the total "mis-clustering" penalties - suppose that we regard $t$ as the "ground truth" cluster label for an instance $\mathbf{x}_i$, then $\psi_{it}$ measures the sum of hinge losses for all incorrect predictions $r$ $(r \neq t)$, which is consistent with the supervised multi-class SVM at a higher level [2]. Eq. 7 is a standard ILP problem with $N \times K$ variables and $N + K$ constraints. We use the GNU Linear Programming Kit (GLPK) to obtain an approximate solution to this problem.

**Updating** $\mathcal{W}$: The next step of learning is the optimization over the model parameters $\mathcal{W}$ (Eq. 5). The learning problem is non-convex and we use the the non-convex bundle optimization solver in [3]. In a nutshell, this method builds a piecewise quadratic approximation to the objective function of Eq. 5 by iteratively adding a linear cutting plane at the current optimum and updating the optimum. Now the key issue is to compute the subgradient $\partial_{\mathbf{w}_t} f_{\mathbf{w}_t}(\mathbf{x}_i)$ for a particular $\mathbf{w}_t$. Let $\mathbf{h}^*_{it}$ be the optimal solution to the inference problem: $\mathbf{h}^*_{it} = \operatorname{argmax}_{\mathbf{h}_{it}} \mathbf{w}_t^\top \Phi(\mathbf{x}_i, \mathbf{h}_{it})$. Then the subgradient can be calculated as $\partial_{\mathbf{w}_t} f_{\mathbf{w}_t}(\mathbf{x}_i) = \Phi(\mathbf{x}_i, \mathbf{h}^*_{it})$. Using the subgradient $\partial_{\mathbf{w}_t} f_{\mathbf{w}_t}(\mathbf{x}_i)$, we optimize Eq. 5 by the algorithm in [3].

## 4  Tag-Based Video Clustering

In this section, we introduce an application of LMMC: tag-based video clustering. Our goal is to jointly learn video clusters and tags in a single framework. We treat tags of a video as latent variables and capture the correlations between clusters and tags. Intuitively, videos with a similar set of tags should be assigned to the same cluster. We assume a separate training dataset consisting of videos with ground-truth tag labels exists, from which we train tag detectors independently. During clustering, we are given a set of new videos without the ground-truth tag labels, and our goal is to assign cluster labels to these videos.

We employ a latent tag model to represent videos. We are particularly interested in tags which describe different aspects of videos. For example, a video from the cluster "feeding animal" (see Figure 1) may be annotated with "dog", "food", "man", etc. Assume we collect all the tags in a set $\mathcal{T}$. For a video being assigned to a particular cluster, we know it could have a number of tags from $\mathcal{T}$ describing its visual content related to the cluster. However, we do not know which tags are present in the video. To address this problem, we associate latent variables to the video to denote the presence and absence of tags.

Formally, given a cluster parameterized by $\mathbf{w}$, we associate a latent variable $\mathbf{h}$ to a video $\mathbf{x}$, where $\mathbf{h} = \{h_t\}_{t\in\mathcal{T}}$ and $h_t \in \{0, 1\}$ is a binary variable denoting the presence/absence of each tag $t$. $h_t = 1$ means $\mathbf{x}$ has the tag $t$, while $h_t = 0$ means $\mathbf{x}$ does not have the tag $t$. Figure 1 shows the latent tag representations of two sample videos. We score the video $\mathbf{x}$ according to the model in Eq. 3: $f_{\mathbf{w}}(\mathbf{x}) = \max_{\mathbf{h}} \mathbf{w}^\top \Phi(\mathbf{x}, \mathbf{h})$, where the potential function $\mathbf{w}^\top \Phi(\mathbf{x}, \mathbf{h})$ is defined as follows:

$$\mathbf{w}^\top \Phi(\mathbf{x}, \mathbf{h}) = \frac{1}{|\mathcal{T}|} \sum_{t\in\mathcal{T}} h_t \cdot \omega_t^\top \phi_t(\mathbf{x}) \tag{8}$$

This potential function measures the compatibility between the video $\mathbf{x}$ and tag $t$ associated with the current cluster. Note that $\mathbf{w} = \{\omega_t\}_{t\in\mathcal{T}}$ are the cluster-specific model parameters, and $\Phi = \{h_t \cdot \phi_t(\mathbf{x})\}_{t\in\mathcal{T}}$ is the feature vector depending on the video $\mathbf{x}$ and its tags $\mathbf{h}$. Here $\phi_t(\mathbf{x}) \in \mathbb{R}^d$ is the feature vector extracted from the video $\mathbf{x}$, and the parameter $\omega_t$ is a template for tag $t$. In our current implementation, instead of keeping $\phi_t(\mathbf{x})$ as a high dimensional vector of video features, we

simply represent it as a scalar score of detecting tag $t$ on $\mathbf{x}$ by a pre-trained binary tag detector. To learn biases between different clusters, we append a constant 1 to make $\phi_t(\mathbf{x})$ two-dimensional.

Now we describe how to infer the latent variable $\mathbf{h}^* = \text{argmax}_{\mathbf{h}} \mathbf{w}^\top \Phi(\mathbf{x}, \mathbf{h})$. As there is no dependency between tags, we can infer each latent variable separately. According to Eq. 8, the term corresponding to tag $t$ is $h_t \cdot \omega_t^\top \phi_t(\mathbf{x})$. Considering that $h_t$ is binary, we set $h_t$ to 1 if $\omega_t^\top \phi_t(\mathbf{x}) > 0$; otherwise, we set $h_t$ to 0.

## 5 Experiments

We evaluate the performance of our method on three standard video datasets: TRECVID MED 11 [19], KTH Actions [26] and UCF Sports [23]. We briefly describe our experimental setup before reporting the experimental results in Section 5.1.

**TRECVID MED 11 dataset** [19]: This dataset contains web videos collected by the Linguistic Data Consortium from various web video hosting sites. There are 15 complex event categories including "board trick", "feeding animal", "landing fish", "wedding ceremony", "woodworking project", "birthday party", "changing tire", "flash mob", "getting vehicle unstuck", "grooming animal", "making sandwich", "parade", "parkour", "repairing appliance", and "sewing project". TRECVID MED 11 has three data collections: Event-Kit, DEV-T and DEV-O. DEV-T and DEV-O are dominated by videos of the null category, i.e. background videos that do not contain the events of interest. Thus, we use the Event-Kit data collection in the experiments. By removing 13 short videos that contain no visual content, we finally have a total of 2,379 videos for clustering.

We use tags that were generated in Vahdat and Mori [28] for the TRECVID MED 11 dataset. Specifically, this dataset includes "judgment files" that contain a short one-sentence description for each video. A sample description is: "A man and a little boy lie on the ground after the boy has fallen off his bike". This sentence provides us with information about presence of objects such as "man", "boy", "ground" and "bike", which could be used as tags. In [28], text analysis tools are employed to extract binary tags based on frequent nouns in the judgment files. Examples of 74 frequent tags used in this work are: "music", "person", "food", "kitchen", "bird", "bike", "car", "street", "boat", "water", etc. The complete list of tags are available on our website.

To train tag detectors, we use the DEV-T and DEV-O videos that belong to the 15 event categories. There are 1675 videos in total. We extract HOG3D descriptors [13] and form a 1,000 word codebook. Each video is then represented by a 1,000-dimensional feature vector. We train a linear SVM for each tag, and predict the detection scores on the Event-Kit videos. To remove biases between tag detectors, we normalize the detection scores by z-score normalization. Note that we make no use of the ground-truth tags on the Event-Kit videos that are to be clustered.

**KTH Actions dataset** [26]: This dataset contains a total of 599 videos of 6 human actions: "walking", "jogging", "running", "boxing", "hand waving", and "hand clapping". Our experiments use all the videos for clustering.

We use Action Bank [24] to generate tags for this dataset. Action Bank has 205 template actions with various action semantics and viewpoints. Randomly selected examples of template actions are: "hula1", "ski5", "clap3", "fence2", "violin6", etc. In our experiments, we treat the template actions as tags. Specifically, on each video and for each template action, we use the set of Action Bank action detection scores collected at different spatiotemporal scales and correlation volumes. We perform max-pooling on the scores to obtain the corresponding tag detection score. Again, for each tag, we normalize the detection scores by z-score normalization.

**UCF Sports dataset** [23]: This dataset consists of 140 videos from 10 action classes: "diving", "golf swinging", "kicking", "lifting", "horse riding", "running", "skating", "swinging (on the pommel horse)", "swinging (at the high bar)", and "walking". We use all the videos for clustering. The tags and tag detection scores are generated from Action Bank, in the same way as KTH Actions.

**Baselines**: To evaluate the efficacy of LMMC, we implement three conventional clustering methods for comparison, including the k-means algorithm (KM), normalized cut (NC) [27], and spectral clustering (SC) [18]. For NC, the implementation and parameter settings are the same as [27], which uses a Gaussian similarity function with all the instances considered as neighbors. For SC, we use a 5-nearest neighborhood graph and set the width of the Gaussian similarity function as the

| | TRECVID MED 11 | | | | KTH Actions | | | | UCF Sports | | | |
|---|---|---|---|---|---|---|---|---|---|---|---|---|
| | PUR | NMI | RI | FM | PUR | NMI | RI | FM | PUR | NMI | RI | FM |
| LMMC | **39.0** | **28.7** | **89.5** | **22.1** | **92.5** | **87.0** | **95.8** | **87.2** | **76.4** | **71.2** | **92.0** | **60.0** |
| MMC | 36.0 | 26.6 | 89.3 | 20.3 | 91.3 | 86.5 | 95.2 | 85.5 | 63.6 | 62.2 | 89.2 | 46.1 |
| SC | 28.6 | 23.6 | 87.1 | 20.3 | 61.0 | 60.8 | 75.6 | 58.2 | 69.9 | 70.8 | 90.6 | 58.1 |
| KM | 27.0 | 23.8 | 85.9 | 20.4 | 64.8 | 60.7 | 84.0 | 60.6 | 63.1 | 66.2 | 87.9 | 58.7 |
| NC | 12.9 | 5.7 | 31.6 | 12.7 | 48.0 | 33.9 | 72.9 | 35.1 | 60.7 | 55.8 | 83.4 | 41.8 |

Table 1: Clustering results (in %) on the three datasets. The figures boldfaced are the best performance among all the compared methods.

average distance over all the 5-nearest neighbors. Note that these three methods do not use latent variable models. Therefore, for a fair comparison with LMMC, they are directly performed on the data where each video is represented by a vector of tag detection scores. We have also tried KM, NC and SC on the 1,000-dimensional HOG3D features. However, the performance is worse and is not reported here. Furthermore, to mitigate the effect of randomness, KM, NC and SC are run 10 times with different initial seeds and the average results are recorded in the experiments.

In order to show the benefits of incorporating latent variables, we further develop a baseline called MMC by replacing the latent variable model $f_{\mathbf{w}}(\mathbf{x})$ in Eq. 4 with a linear model $\mathbf{w}^\top \mathbf{x}$. This is equivalent to running an ordinary maximum margin clustering algorithm on the video data represented by tag detection scores. For a fair comparison, we use the same solver for learning MMC and LMMC. The trade-off parameter $C$ in Eq. 4 is selected as the best from the range $\{10^1, 10^2, 10^3\}$. The lower bound and upper bounds of the cluster-balance constraint (i.e. $L$ and $U$ in Eq. 4) are set as $0.9\frac{N}{K}$ and $1.1\frac{N}{K}$ respectively to enforce balanced clusters.

**Performance measures**: Following the convention of maximum margin clustering [32, 33, 29, 37, 38, 16, 6], we set the number of clusters to be the ground-truth number of classes for all the compared methods. The clustering quality is evaluated by four standard measurements including purity (PUR) [32], normalized mutual information (NMI) [15], Rand index (RI) [21] and balanced F-measure (FM). They are employed to assess different aspects of a given clustering: PUR measures the accuracy of the dominating class in each cluster; NMI is from the information-theoretic perspective and calculates the mutual dependence of the predicted clustering and the ground-truth partitions; RI evaluates true positives within clusters and true negatives between clusters; and FM considers both precision and recall. The higher the four measures, the better the performance.

## 5.1  Results

The clustering results are listed in Table 1. It shows that LMMC consistently outperforms the MMC baseline and conventional clustering methods on all three datasets. Specifically, by incorporating latent variables, LMMC improves the MMC baseline by 3% on TRECVID MED 11, 1% on KTH Actions, and 13% on UCF Sports respectively, in terms of PUR. This demonstrates that learning the latent presence and absence of tags can exploit rich representations of videos, and boost clustering performance. Moreover, LMMC performs better than the three conventional methods, SC, KM and NC, showing the efficacy of the proposed LMMC framework for unsupervised data clustering.

Note that MMC runs on the same non-latent representation as the three conventional methods, SC, KM and NC. However, MMC outperforms them on the two largest datasets, TRECVID MED 11 and KTH Actions, and is comparable with them on UCF Sports. This provides evidence for the effectiveness of maximum margin clustering as well as the proposed alternating descent algorithm for optimizing the non-convex objective.

**Visualization**: We select four clusters from TRECVID MED 11, and visualize the results in Figure 2. Please refer to the caption for more details.

## 6  Conclusion

We have presented a latent maximum margin framework for unsupervised clustering. By representing instances with latent variables, our method features the ability to exploit the unobserved information embedded in data. We formulate our framework by large margin learning, and an alter-

| **Cluster**: woodworking project | **Cluster**: birthday party |
|---|---|
| 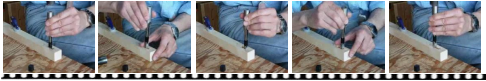 ✔ |  ✔ |
| **Tags**: piece, wood, machine, lady, indoors, man, kitchen, baby | **Tags**: party, birthday, restaurant, couple, wedding ceremony, wedding, ceremony, indoors |
|  ✔ |  ✔ |
| **Tags**: piece, man, wood, baby, hand, machine, lady, kitchen | **Tags**: birthday, party, restaurant, family, child, wedding ceremony, wedding, couple |
|  ✔ |  ✔ |
| **Tags**: wood, piece, baby, indoors, hand, man, lady, bike | **Tags**: party, birthday, restaurant, child, family, wedding ceremony, chicken, couple |
| **Cluster**: parade | **Cluster**: landing fish |
|  ✘ |  ✘ |
| **Tags**: city, day, year, Chinese, Christmas, people, lot, group | **Tags**: fish, fishing, boat, man, beach, line, water, woman |
|  ✔ |  ✔ |
| **Tags**: day, street, lot, Chinese, year, line, Christmas, dance | **Tags**: boat, beach, fish, man, men, group, water, woman |
|  ✔ |  ✔ |
| **Tags**: street, day, lot, Chinese, line, year, dancing, dance | **Tags**: fish, beach, boat, men, man, chicken, truck, move |

Figure 2: Four sample clusters from TRECVID MED 11. We label each cluster by the dominating video class, e.g. "woodworking project", "parade", and visualize the top-3 scored videos. A "✔" sign indicates that the video label is consistent with the cluster label; otherwise, a "✘" sign is used. The two "mis-clustered" videos are on "parkour" (left) and "feeding animal" (right). Below each video, we show the top eight inferred tags sorted by the potential calculated from Eq. 8.

nating descent algorithm is developed to solve the resultant non-convex objective. We instantiate our framework with tag-based video clustering, where each video is represented by a latent tag model with latent presence and absence of video tags. Our experiments conducted on three standard video datasets validate the efficacy of the proposed framework. We believe our solution is general enough to be applied in other applications with latent representations, e.g. video clustering with latent key segments, image clustering with latent region-of-interest, etc. It would also be interesting to extend our framework to semi-supervised learning by assuming a training set of labeled instances.

### Acknowledgments

This work was supported by a Google Research Award, NSERC, and the Intelligence Advanced Research Projects Activity (IARPA) via Department of Interior National Business Center contract number D11PC20069. The U.S. Government is authorized to reproduce and distribute reprints for Governmental purposes notwithstanding any copyright annotation thereon. Disclaimer: The views and conclusions contained herein are those of the authors and should not be interpreted as necessarily representing the official policies or endorsements, either expressed or implied, of IARPA, DOI/NBC, or the U.S. Government.

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
