[Reviews · NeurIPS 2013]

Submitted by Assigned_Reviewer_4

This paper improves maximum margin clustering method by introducing latent variables into the model. The latent variables are able to capture the tag information embedded in the data. Overall, this is a decent paper. The idea is interesting, and experiments demonstrate the value of the proposed method.
- Just to clarify, is the number of clusters (K) assumed to be given?
- The authors demonstrate the proposed method on only one application – Tag-based video clustering. It would be nice to apply this method to other applications.
- The proposed model is generally intractable (the minimization of y). Therefore, an approximation is required. The authors formulate the problem as an ILP and solve it approximately by GLPK. It would be nice to check the influence of this approximation using some small data.
Summary: Overall, this is a decent paper. The idea is interesting, and experiments demonstrate the value of the proposed method. However, it would be more interesting if the method can be applied to more than one application domains.

Submitted by Assigned_Reviewer_6

The authors provide a model and algorithm for unsupervised clustering. The method extends max-margin clustering (unsupervised SVM) by adding latent variables for the classifier associated with each cluster. During inference, these variables are maximized.

The intuition behind adding this new degree of freedom is that it would allow for heterogenous categories, in which different aspects of the category are indexed by the latent variables. This is the case for supervised latent SVMs. However, in the unsupervised setting, there is likely to be an identification problem: When no labels are available, how does the algorithm "know" which clusters are categories and which are sub-aspects of the same category?

The experimental justification for adding latent variables is somewhat lacking. The performance difference between MMC and Latent MMC is rather modest. Furthermore, no error bars are present to tell whether or not this difference is statistically significant over repeated runs of the algorithms. Do the latent variables have any qualitative meaning, e.g. do they actually identify different aspects of a category? Latent MMC has more degrees of freedom than MMC. What if we solve MMC in a space of higher dimension by using a kernel? Would the performance gap decrease? This paper only compares Latent MMC against their own MMC algorithm. What about comparing to other established algorithms for MMC?

Finally, the proposed algorithm is somewhat unsatisfying since no convergence guarantees are made. It's clear that the algorithm will converge to some solution, but nothing is said about the nature of this solution. It's understood that clustering is generally a non-convex optimization problem so global guarantees can't be given, but most established clustering algorithms provide a guarantee to convergence at a local optima.
Summary: An interesting problem, but the advantages of this method haven't been demonstrated sufficiently enough to justify publication in NIPS.

Submitted by Assigned_Reviewer_8

Summary:

This work proposes an extension to the maximum margin clustering (MMC) method that introduces latent variables. The motivation for adding latent variables is that they can model additional data semantics, resulting in better final clusters. The authors introduce a latent MMC (LMMC) objective, state how to optimize it, and then apply it to the task of video clustering. For this task, the latent variables are tag words, and the affinity of a video for a tag is given by a pre-trained binary tag detector. Experiments show that LMMC consistently, and sometimes substantially, beats several reasonable baselines.

Comments:

Clarity: The paper is clear and well-written. The related work section is extensive and the experimental details are abundant. Noticed just a few typos:

1) Line 83: The "and combines them" should be "and combine them".

2) Line 187: The "Balanced clusters are constrained by the last constraint" should be something like "Cluster balance is enforced by the last constraint".

Originality: Adding latent variables to MMC is relatively novel. The changes to the MMC objective and optimization are not too substantial, but nonetheless represent something new. Very recent work that somewhat overlaps with this work is that of Yang Wang and Liangliang Cao, "Discovering Latent Clusters from Geotagged Beach Images" (MMM 2013). As this work also deals with adding latent variables, though in a different manner, it would be good if the authors could indicate the ways in which their LMMC method differs.

Significance: The empirical validation presented in this work is strong enough to indicate that, at a minimum, LMMC is significant for video clustering. Most likely it is also applicable to many other clustering tasks, which frequently arise in the machine learning community. However, it is hard to be sure from the current experiments that the latent variables actually correspond to coherent sub-aspects of a category.

Quality: This paper is of good quality. Its math and experiments seem relatively sound. While the delta from MMC is not very large in the statement of the objective and the exposition of the optimization, the experiments clearly show that the new method represents a step up accuracy-wise. One suggestion --- it would be interesting to see some runtime comparisons in Table 1.
Summary: This work modifies maximum margin clustering by adding latent variables. The change to MMC's math seems small, but the experiments show that adding latent variables has a significant effect on the accuracy (though whether the latent variables truly correspond to coherent sub-aspects of a category is debatable).

Submitted by Assigned_Reviewer_9

This paper develops a method for clustering and inferring about latent variables associated with each instance simultaneously. Their method builds on the maximum margin clustering (MMC) method by incorporating the latent variable model. They formalize the the clustering objective into a non-convex optimization problem, where the latent variable assignment, the labeling of the cluster, and the linear discriminant function parameters are iteratively optimized, using an alternating descent algorithm.

The paper is well developed and has good theoretical derivation. However, it seems to be not so clear in explaining the reason for an extension from the MMC to Latent MMC. Besides the application-specific requirement for inferring about latent variables, for example, the tagging task in their experiment, by incorporating the latent variables into the optimization objective, does it provide more information for clustering? Or does the clustering result provide information for inference on latent variables? Or both? Could the authors provide more general utilization of their framework? Could the latent variable framework be more useful when we are in a semi-supervised setting where we have some way of comparing the generated latent variable values with some ground truth values?

Also, it seems to me that the optimization problem formulated in the paper should take a substantial amount of time to solve. In the Experiment section, the paper compares their clustering result with the others in terms of correctness. Could the authors provide some comparison on the time complexity as well?
Summary: The paper is well-organized and developed but seems to need more convincing support for the adoption of their new model.
Author Feedback

Author rebuttal: Reviewer 4

As stated in lines 356-358, we fix the number of cluster to be the ground-truth number of classes on each dataset. This follows the convention of maximum margin clustering in measuring performance [31, 32, 29, 26, 27, 16, 6].

Though we focus on tag-based video clustering, the LMMC framework is generic and could be applied to various applications with latent representations, e.g. video clustering with latent key segments, image clustering with latent region-of-interest, object clustering with latent scale/location configurations, etc.


Reviewer 6

We totally agree that latent variables can be used to represent sub-categories (e.g. [33]). However, latent variables are not limited to sub-categorizations. For example, DPMs [5] model latent scale/location configurations together with latent sub-categories for object detection. Izadinia and Shah [10] model low-level events (e.g. people dancing, animal eating) as latent variables to recognize complex video events (e.g. wedding ceremony, grooming animal). In our experiments, we cluster videos by considering the latent presence/absence of semantic tags -- each latent variable helps to determine whether a video has a specific tag or not. For clustering, videos with similar semantic tags can be grouped together to form a cluster. Note that our latent variables do not specifically identify sub-categories. However, we emphasize that our solution is generic and can be used to cluster data with sub-category based latent variables.

The optimizations of LMMC and MMC are deterministic, so the results will remain the same over repeated runs. We show experimentally that LMMC outperforms MMC on three standard datasets in terms of all the four popularly-used performance measures.

The latent variables generally have semantic meanings. For example, in our video clustering application, the latent variables determine the presence/absence of semantic tags on each video (see Figure 2 for some qualitative evaluations).

We compare LMMC with MMC using linear models for both, for a fair comparison. We agree that kernelizing MMC could potentially boost its performance, and would likely also boost LMMC's performance.

As for convergence, LMMC involves optimizing a discrete Y and continuous W. The learning of LMMC consists of two alternating steps: risk evaluation and weight updating. In the weight updating step, once we have fixed latent variables H and cluster labeling assignment Y, the model parameters W are guaranteed to converge to a local optima [3]. Hence, we are guaranteed convergence in a weak sense, to a local optimum for W for the discrete Y. In the risk evaluation step, the inference of Y is an NP-hard problem. Thus, in practice, we develop an approximate solution based on ILP. This approximate inference can enforce an improvement in the learning objective with respect to the previous Y, hence convergence (in the discrete space of Y, with a W that is a local optimum for this Y).


Reviewer 8

Thanks for pointing out the typos and the missing reference. We will fix them if our paper is accepted.

To address the reviewer’s concern, we collect the runtime results (in seconds) for all the compared methods:

---------- TRECVID ------- KTH ------- UCF

LMMC ---- 297.270 ----- 9.060 ----- 8.480

MMC ---- 5209.510 ---- 15.820 ---- 82.010

SC ---------- 1.215 ----- 0.208 ----- 0.065

KM ---------- 0.112 ----- 0.016 ----- 0.004

NC ---------- 2.550 ----- 0.169 ----- 0.078

On all three datasets, LMMC and MMC take more time than the three conventional clustering methods. This is because LMMC and MMC need a training process to learn model parameters. Besides, it is interesting to note that LMMC costs less time than MMC. The reason is that learning MMC involves all the tags on all the videos, but learning LMMC only deals with a sparse set of latent tags.


Reviewer 9

The reason for extending MMC to LMMC is to integrate additional sources of semantics into the model in order to achieve better clustering results. See our application -- we improve the video clustering results by predicting which tags are associated with each video (latent variables) during inference. The two steps are coupled in a joint optimization framework. Though we focus on a particular application, the proposed framework is generic -- one can incorporate varied sources of information into the framework through latent variables. The proposed LMMC can be directly extended to a semi-supervised setting by fixing some of the latent variables to the ground-truth values during learning.

Please refer to the response to Reviewer 8 for a detailed comparison of the runtime results.